# Articular Cartilage Regeneration by Hyaline Chondrocytes: A Case Study in Equine Model and Outcomes

**DOI:** 10.3390/biomedicines11061602

**Published:** 2023-05-31

**Authors:** Fernando Canonici, Cristiano Cocumelli, Antonella Cersini, Daniele Marcoccia, Alessia Zepparoni, Annalisa Altigeri, Daniela Caciolo, Cristina Roncoroni, Valentina Monteleone, Elisa Innocenzi, Cristian Alimonti, Paola Ghisellini, Cristina Rando, Eugenia Pechkova, Roberto Eggenhöffner, Maria Teresa Scicluna, Katia Barbaro

**Affiliations:** 1Equine Practice s.r.l., Campagnano, Strada Valle del Baccano 80, 00063 Rome, Italy; fernandocanonici@gmail.com; 2Istituto Zooprofilattico Sperimentale del Lazio e della Toscana “M. Aleandri”, Via Appia Nuova 1411, 00178 Rome, Italy; cristiano.cocumelli@izslt.it (C.C.); antonella.cersini@izslt.it (A.C.); daniele.marcoccia@izslt.it (D.M.); alessia.zepparoni@izslt.it (A.Z.); annalisa.altigeri@izslt.it (A.A.); daniela.caciolo@izslt.it (D.C.); cristina.roncoroni@izslt.it (C.R.); valentina.monteleone-esterno@izslt.it (V.M.); elisa.innocenzi-esterno@izslt.it (E.I.); teresa.scicluna@izslt.it (M.T.S.); 3Department of Surgical Sciences and Integrated Diagnostics (DISC), University of Genoa, 16132 Genoa, Italy; paola.ghisellini@unige.it (P.G.); cristina.rando@unige.it (C.R.); roberto.eggenhoffner@unige.it (R.E.); 4Consorzio Interuniversitario INBB, Viale delle Medaglie d’Oro 305, 00136 Rome, Italy; eugenia.pechkova@gmail.com; 5Department of Experimental Medicine (DIMES), University of Genoa, 16132 Genoa, Italy

**Keywords:** articular cartilage repair, regenerative medicine, horse, hyaline cartilage

## Abstract

Cartilage injury defects in animals and humans result in the development of osteoarthritis and the progression of joint deterioration. Cell isolation from equine hyaline cartilage and evaluation of their ability to repair equine joint cartilage injuries establish a new experimental protocol for an alternative approach to osteochondral lesions treatment. Chondrocytes (CCs), isolated from the autologous cartilage of the trachea, grown in the laboratory, and subsequently arthroscopically implanted into the lesion site, were used to regenerate a chondral lesion of the carpal joint of a horse. Biopsies of the treated cartilage taken after 8 and 13 months of implantation for histological and immunohistochemical evaluation of the tissue demonstrate that the tissue was still immature 8 months after implantation, while at 13 months it was organized almost similarly to the original hyaline cartilage. Finally, a tissue perfectly comparable to native articular cartilage was detected 24 months after implantation. Histological investigations demonstrate the progressive maturation of the hyaline cartilage at the site of the lesion. The hyaline type of tracheal cartilage, used as a source of CCs, allows for the repair of joint cartilage injuries through the neosynthesis of hyaline cartilage that presents characteristics identical to the articular cartilage of the original tissue.

## 1. Introduction

Over the last few years, issues related to bone and tissue reconstruction techniques have assumed increasing importance, both for their benefits in animal and human applications. Among these, the treatment of cartilage lesions in the orthopedic field is of great importance since the articular cartilage has a limited regenerative capacity due to the absence of vascularization and the low quantity and low replicative activity of the specialized cells present. Furthermore, it is known that joint cartilage damage, if not properly treated, can easily evolve into arthritic-type degeneration. To date, different techniques are used for the treatment of this type of injury: debridement, drilling of the subchondral bone, periosteal transplantation and perichondral reconstruction, and more recently, transplantation of mesenchymal stem cells from bone marrow and adipose tissue. These approaches have not always proved to be effective, leading, in some cases, to the formation of fibrocartilaginous tissue characterized by minor characteristics.

In joint space, the mainly present cartilage is of the hyaline type; the latter is characterized by a low number of chondrocytes (CCs) immersed in an abundant extracellular matrix (ECM) and due to its poor vascularization, it has limited repair capacity [1]. Defects in joint cartilage can induce debilitating degenerative joint diseases such as osteoarthritis (OA), which represent one of the most common causes of lameness and can compromise the quality of life and performance of animals, particularly sport horses [2,3,4]. Joint cartilage injuries that exceed a dimension of 15 mm^2^ turn out to be particularly critical because they cannot heal spontaneously [5] and can quickly evolve into a type of degenerative arthritis [3]. For the treatment of these lesions, among those commonly used, we consider two main categories of techniques: local surgical procedures and cellular tissue transplantation techniques [5,6,7].

The surgical cartilage treatments are carried out for the two-fold purpose of promoting the repair of the damage and stimulating new cartilage growth. These approaches were grouped into three main categories: (i) palliative (arthroscopic debridement and lavage), (ii) reparative (marrow stimulation techniques) and (iii) restorative (osteochondral grafting, autologous CC implantation (ACI) [7,8,9,10]. Arthroscopic debridement and joint lavage are palliative treatments that are carried out together with the removal of free bodies in the joint and limited excision of osteophytes to decrease pain and improve joint function [7,11]. Instead, bone marrow stimulation techniques (BSTs) are surgical treatments carried out for symptomatic articular cartilage defects [7,12]. BSTs rely on the penetration of the subchondral bone plate at the base of the cartilage injury [5,7]. The use of autologous CCs is among the most compelling cellular regenerative medicine techniques used for cartilage repair due to the intrinsic potential of these cells for recreating hyaline cartilage [13]. However, abrasion arthroplasty is used to treat small flaws due to its ability to remove the damaged cartilage and reach the subchondral bone [4,14]. The applications of these techniques for the repair of cartilage damage can implicate the formation of fibrocartilaginous rather than hyaline tissue, which may present reduced biomechanical and functional characteristics, leading to the formation of joints vulnerable to shear stresses and prone to breaking over time [9,15]. Cartilage repair studies are conducted using different cellular sources, such as adipose tissue, bone marrow, blood, synovial fluid and different techniques of cell collection [16,17]. Moreover, the use of scaffolds, growth factors and cellular paths, as well as the number of cells used, turn out to be extremely heterogeneous due to the lack of correlations between preclinical data on animals and humans [18,19]. Therefore, the development of new regeneration methods aims at producing a tissue more similar to hyaline articular cartilage with suitable biomechanical properties that could guarantee an optimal recovery of the injuries that cannot be repaired on their own [9,15,20]. The present study describes the application of regenerative therapy to damaged articular cartilage in an athletic horse, with a two-year follow-up of the clinical performance and rehabilitation up to the return to racing, corroborated by ancillary medical examinations, periodic arthroscopy, and biopsies of the repairing cartilage. Cell therapy with chondrocytes isolated from tracheal cartilage could be a valid alternative to traditional therapies for the treatment of lesions affecting the articular cartilage, not only in horses, but also in other animals and humans.

## 2. Materials and Methods

For this study, a three-year-old female thoroughbred employed in flat horse races was recruited. The horse was affected by a chronic chip fracture of the right antebrachiocarpal joint, with a lesion on the surface cartilaginous distal of the radio, in a dorsolateral position, with an evident cartilage flap of approximately 12 mm × 7 mm × 6 mm in size. The horse’s medical practices were carried out at the veterinary hospital (Equine Practice, Campagnano, Rome, Italy), after the authorization of the owner.

### 2.1. Hyaline Cartilage Sample Preparation

A sample of autologous tracheal hyaline cartilage was collected after the sedation of the horse by using 4 mg of detomidine. Local anesthesia was injected under the skin of the ventral and axial aspects of the proximal third of the neck by using 10 mL of 2% mepivacaine. After trimming and aseptic preparation of the area, a vertical incision of 7–8 cm in the skin was performed, the two bellies of the sternohyoid muscle were separated, and the trachea was exposed. A sample of about 1 cm^2^ of a tracheal ring was collected without penetrating the tracheal lumen to avoid any risk of contamination, immediately immersed in αMEM (Gibco, Billings, MT, USA), and transferred for laboratory processing.

### 2.2. CCs Isolation and Growth

After careful removal of the perichondrium from the biopsies, the tissue was cut into small pieces, and subjected to enzymatic digestion with a solution of collagenase IA at 0.075% (Gibco, Billings, MT, USA) for 1 h at 37 °C and subsequently treated with trypsin-EDTA at 0.05% (Gibco, Billings, MT, USA) for 1 h at 37 °C. At the end of the digestion, the sample obtained was centrifuged at 1200× *g* for 8 min, and the pellet was washed three times with αMEM containing 10% fetal bovine serum (FBS, Gibco, Billings, MT, USA). The collected cells were resuspended in αMEM containing 10% FBS, 100 U/mL penicillin (Sigma, Burlington, MA, USA), 100 μg/mL streptomycin (Sigma, Burlington, MA, USA) and 2.50 μg/mL amphotericin B (Sigma, Burlington, MA, USA), and seeded in 75 cm^2^ plastic flasks at 37 °C in 5% CO_2_.

Isolated CCs were observed on a daily basis by using an inverted optical microscope Nikon Eclipse TE2000-U (Nikon Minato, Tokyo, Japan) to evaluate over time both the growth and the morphological changes of the cell population. After 24 h, growth medium was changed to remove nonadherent cells.

At 75–80% cellular confluence, a mixture of trypsin at 0.05% and EDTA at 0.01% was added. The enzymatic action of trypsin was interrupted by using the αMEM culture medium as FBS at 10%, and subsequently the cell cultures were incubated under the same conditions previously described. The successive steps were handled with the same procedure; before reaching the 75–80% confluent monolayer, the culture medium was changed every two days.

#### Differentiation Potential of CCs

The differentiation potential of the chondrocytes towards the osteogenic, adipogenic and chondrogenic lineages was evaluated at the 2nd passage in culture. The cells were enzymatically detached and distributed in wells of 24 plates at 3 different concentrations: 50,000 cells/mL, 35,000 cells/mL and 20,000 cells/mL and each lineage was evaluated in triplicate. The cells were placed in an incubator at 37 °C in a humidified atmosphere of 5% CO_2_ for 24 h. The cultures were then stimulated with the appropriate differentiation medium under the conditions discussed in the following sections a–c.

osteogenic differentiation

Cultures were stimulated for 2 weeks in αMEM containing 10% FCS supplemented with 50 µg/mL ascorbic acid (Sigma, Burlington, MA, USA), 10 mM ß-glycerophosphate (Sigma, Burlington, MA, USA), 10^−7^M dexamethasone (Sigma, Burlington, MA, USA). The negative control was represented by chondrocytes grown in αMEM containing 10% FCS.

2.adipogenic differentiation:

Cultures were stimulated for 2 weeks in αMEM containing 10% FCS supplemented with 1 µM dexamethasone, 10 µg/mL insulin (Sigma, Burlington, MA, USA), 0.2 mM indomethacin (Sigma, Burlington, MA, USA) and 0.5 mM 3-isobutyl-methyl-xanthine (Sigma, Burlington, MA, USA). The negative control was represented by chondrocytes grown in αMEM containing 10% FCS.

3.chondrogenic differentiation:

Cultures were stimulated for 2 weeks with αMEM containing 1% FCS supplemented with 0.1 µM dexamethasone, 6.25 µg/mL insulin, 50 nM ascorbic acid and 10 ng/mL TGFα. The negative control was represented by chondrocytes grown in αMEM containing 10% FCS.

### 2.3. CCs Suspension Implantation

The implant of the stem cell suspension was performed through arthroscopy of the antebrachiocarpal joint under general anesthesia with the horse in dorsal recumbency, as described by Martin and Mc Ilwraith [11].

After the removal of the osteochondral fragment from the distolateral aspect of the radius and debridement of the surrounding area, two holes of 2.5 mm in diameter and 5 mm in depth were drilled in the exposed subchondral bone and threaded with a 3.5 mm tap, intending to augment the grip of the implant. In the first hole, a volume of 1.5 mL of autologous fibrinogen “home-made” concentrate [21] with 30 × 10⁶ CCs was injected and agglutinated in situ with 10 µL of human thrombin (Tissucol, Baxter Deerfield, Illinois, USA). The second hole was not treated for tissue regeneration and, for this reason, was regarded as a control.

The first surgical part was performed with liquid distension of the articular space, whereas the part of implantation was performed with gas distention [22], to avoid the dilution of the cell suspension. Once the consistency of the implant could be established, the skin incisions were sutured, keeping the limb aseptically bandaged.

### 2.4. Arthroscopy Follow-Up and Biopsies

The clinical follow-up included the clinical examination and the arthroscopy exam with biopsies, performed at 8 and 13, and 24 months for the surgical treatment. All the bioptic samples were routinely processed for histology after 24 h of fixation in buffered formalin. The slides, 4 µm thick, were stained with hematoxylin, eosin, and toluidine blue. Further sections were processed through immunohistochemistry (IHC) to detect the antigen expression of type II collagen. Briefly, after peroxidase blocking, 3% H_2_O_2_ in PBS was used to block cartilage peroxidase for 10 min.

Slides were incubated at 37 °C for 1 h with 2.5% hyaluronidase in 7.4 pH PBS in a wet chamber, then covered with normal goat serum diluted 1:10 in bovine serum albumin BSA, (Sigma, Burlington, MA, USA). Afterwards, the slides were incubated overnight with the polyclonal anti-COL2A1 antibody (Collagen II 1 Alpha Antibody, LSBIO LS-C354627 Lynnwood, WA, USA, 1:100 dilution). Further steps followed the manufacturer’s instructions (Dako EnVision, Carpinteria, CA, USA).

## 3. Results

### 3.1. Morphological Characterization of Cultured Chondrocytes

The CCs obtained after digestion and cultures of the hyaline cartilage biopsy were present in large quantities and displayed high replicative power. Moreover, the CCs population showed the ability to adhere to the plastic substrate of the culture flask.

At the first step, these cells were small and polygonal with the roundish morphology typical of mature CCs, and in some areas, they appeared to be superimposed in several layers. In particular, at the third and fourth passages, the cells looked larger and broader and showed a fibroblast-like morphology, an index of dedifferentiation. The cells acquired the ability to rapidly and intensively multiply, and this characteristic persisted for all the following phases. Most likely, such an ability depends on the dedifferentiation of these cells in vitro. In fact, the in vitro amplification of isolated cells provides a cellular population with a much more homogeneous morphology than that obtained with isolation alone [23].

#### Evaluation of Differentiation

The population of chondrocytes isolated from the trachea was evaluated from the point of view of multipotentiality by the study of their ability to differentiate in vitro in osteogenic lines, adipogenic and chondrogenic.

The differentiative capacity of the chondrocyte population in the three lineages was highlighted after appropriate stimulation, both by staining the monolayer cells or by the detection of specific RNA. 

The chondrocytes stimulated by osteogenic differentiation showed a change in morphology and extracellular matrix mineralization, highlighted in red with the Alizarin S. color. The microRNAs detected and specific for osteogenesis in mammals are ALP (alkaline phosphatase) and Runx2.

In chondrogenic differentiation, deposits of intra- and extracellular glycosaminoglycans were observed, highlighted in blue with Alcian staining. MicroRNAs tested, measured and related to chondrogenesis in mammals are: AGG (aggrecan), Col-II (type II collagen) and Sox9 (sex-determining region Y-box).

Adipogenic differentiation was observed in rounded cells with lipid vesicles in the cytoplasm, highlighted in red by staining with Oil Red O. The tested microRNAs that were detected and are specific to adipogenesis in mammals are PPARγ2 (peroxisome proliferator-activated receptor γ2).

After osteogenic differentiation, calcium deposits were highlighted by Alizarin Red S staining (Sigma, Burlington, MA, USA). To perform the staining, the cells were fixed for one hour at room temperature in a 70% ethanol solution; subsequently, they were washed with laboratory-grade water and covered with a 2% solution of Alizarin Red S for 30 min. After four washes with laboratory-grade water, the calcium deposits turn orange-red, as shown in Figure 1a.

After adipogenic differentiation, lipid vacuoles accumulated in the cytoplasm and were highlighted by Oil Red O (Sigma, Burlington, MA, USA) staining. After 3 weeks, the cells were fixed with a 10% formalin solution for 1 h at room temperature, then washed with reagent-grade distilled water and subsequently with 60% isopropanol for 5 min and then stained with a solution of Oil Red O (0.3% in 60% isopropanol) for 5 min. After this time, the dye solution was removed, and the cells were washed with reagent-grade distilled water, as shown in Figure 1b. After the chondrogenic differentiation, the intra- and extracellular glycosaminoglycans were highlighted by staining with Alcian Blue (Sigma, Burlington, MA, USA), as shown in Figure 1c. After 3 weeks, the cells were fixed with a 10% formalin solution for 1 h at room temperature, then washed with reagent-grade distilled water and stained with a solution of Alcian Blue in Acetic Acid (Sigma, Burlington, MA, USA) (1% in a 3% solution of acetic acid; pH 2.5) for 15 min at room temperature. After this time, the dye solution was removed, and the cells were washed three times with 3% acetic acid.

### 3.2. Arthroscopic Evaluation of the Lesion Site

In Figure 2, the images of the first arthroscopy exam that identified the site of the lesion are reported. The chondral lesion shows an evident cartilaginous fragment, which was removed during surgery (Figure 2a). In Figure 2b, the articular cartilage of the operative site prepared for the insertion of autologous chondrocytes is shown.

At 8 months after implant, the direct visualization of the operated site by means of arthroscopy revealed the presence of fibrocartilage, with several sites being more compact than others. In fact, the treated site results in exuberant fibrocartilage toward the dorsal aspect, adherent to the capsule (Figure 3a).

Indeed, in Figure 3b,c, the more palmar part of the treated joint is shown, in which thinned cartilage and an island of tissue more similar to hyaline cartilage are evident.

After 13 months from surgery, in arthroscopy, the dorsal face of the treated cartilage was covered by thicker fibrocartilage, while the central part presented a more compact area (Figure 4).

A final control was performed 24 months after surgery (Figure 5), and tissue entirely comparable to the native articular cartilage was observed: the site of the lesion prepared for the implant of the chondrocytes at the first surgery (Figure 6a) and the same site after 24 months (Figure 6b).

### 3.3. Microscopic Evaluation of the Lesion Site

Bioptic samples taken at 8 months post-surgery showed in both samples a deep layer of hyaline cartilage adherent to the subchondral bone, covered by a proliferation of immature tissue; in the treated site, proliferating cells are organized in small clusters (interpreted as proliferating chondrocytes that do not express type II collagen antigens) (Figure 7). In the untreated site (control), the overlying proliferating cells have round to elongated single nuclei embedded in a high amount of fibrous matrix and only a few scattered clusters of nuclei, which do not express type II collagen antigens.

At 13 months post-surgery, chondrocytes in the treated lesion progressively isolate from each other, have rounder nuclei, and are surrounded by a high amount of homogeneous matrix that expresses a slight patchy positive reaction to anti-COL2A1, mimicking mature hyaline cartilage (Figure 8a).

The untreated sample cells keep growing in a fascicular pattern and are supported by a clearly fibrous, slightly eosinophilic matrix, resembling mature fibrous connective tissue; in this sample, expression of COL2A1 cannot be detected (Figure 8b,c).

After twenty-four months from the surgery, cartilage morphology of the hyaline type was present throughout the biopsy sampled up to the surface of the treated site. The implanted and proliferated cells have differentiated towards chondrocytes with the ability to produce the collagen matrix typical of hyaline cartilage (expression of type 2 collagen), as shown in Figure 9.

### 3.4. Clinical Evaluation

The clinical case under study presented a chronic parcellar fracture of the dorsodistal aspect of the right radius. During the first arthroscopy, the lesion site was cleaned and prepared for the inoculation of autologous CCs. After the surgery, the horse was at rest in the box for 4 weeks, walked at a walking pace for 4 weeks and ran in a small paddock for 4 weeks, demonstrating a progressive recovery.

After the second arthroscopy, 8 months after the CCS inoculation, the horse showed a recovery from the surgery. After 13 months after surgery, at clinical examination, the horse had moderate distension of the radiocarpal joint while a synovial sheath of the dorsal extensor muscle of the finger appeared. At twenty-four months after the first surgery, the horse was sound, without response to the flexion test and without distention of the antebrachiocarpal joint or of the digital sheath of the dorsal digital extensor tendon.

## 4. Discussion and Conclusions

Cartilage defects due to injury are seen in animals and humans and commonly result in the development of osteoarthritis and the progression of joint deterioration. The natural healing of cartilage injuries promotes, in most cases, the formation of fibrocartilage, which is histochemically and biomechanically inferior to normal hyaline cartilage [9,24]. The ability of damaged articular cartilage to regenerate normal hyaline cartilage is limited because of two main factors: the absence of a vascular response and the relative absence of an undifferentiated cell population to respond to injury. In addition, the techniques commonly used for these pathologies, such as subchondral perforation or microfractures [14,25], are ineffective if used alone, lead to osteoarthritis [7] and are ineffective for significant osteochondral defects.

To confirm the multi-lineage differentiation capabilities of equine tracheal CCs, the cell cultures were treated with osteogenic, adipogenic and chondrogenic inducers, and after a few weeks of stimulation, it was possible to demonstrate their differentiation in this regard. Specifically, differentiation in an osteogenic sense was highlighted by observing intracellular calcium deposits, differentiation in an adipogenic sense by ascertaining the presence of lipid vacuoles and differentiation in a chondrogenic sense by observing intra- and extracellular deposits of glycosaminoglycans.

The subject in the study presented a 12-mm lesion in the form of a fracture chip of the carpus joint, far over the critical size of the cartilage defect in the equine species, set in the range of 4–9 mm. In this context, regenerative medicine based on cell therapy represents the most promising medical approach for cartilaginous lesions compared to traditional procedures.

In our study, cell therapy was performed with autologous CCs isolated from the tracheal cartilage of the horse. The trachea represents a suitable anatomical region for collecting hyaline cartilage and subsequently amplifying CCs [24]. The horse represents an excellent animal model due to the frequent appearance of arthritic phenomena of traumatic origin [2]. Furthermore, the pathophysiology of equine osteoarthritis and the block peroxidase of the knee cartilage are similar to those of humans [2,26,27].

Furthermore, the results of our work reinforce the expectations reported in a recent review [1] regarding the formation of hyaline cartilage, its coherent morphology, and the ability to successfully treat this type of injury through the use of tracheal stem cells.

The implant of the autologous amplified CCs can be easily performed under arthroscopic control, and arthroscopy clinical follow-up was carried out to evaluate the safety, efficacy, and clinical outcome. The histological techniques employed allowed us to follow articular cartilage recovery, evaluating the changes in morphology and cellular disposition and the extracellular matrix’s biochemical composition (type II collagen, Coll II) up to two years after surgery. After two follow-up years, histological and IHC analysis of the regenerated cartilage biopsies showed extremely satisfactory clinical and anatomical results. The biopsies and histological examination carried out at 8, 13 and 24 months from the implant illustrate the healing, as disclosed by the formation of new cartilaginous tissue, with a notable presence of type II collagen, typical of hyaline cartilage, and by the high cellularity of CCs in clusters immersed in an abundant extracellular matrix. The horse was chosen as an animal model for the re ration of cartilage joints, checked through arthroscopy, clinical follow-ups, biopsies and as well as therapies (surgical and medical), that is, following a procedure similar to that adopted for humans.

Cell therapy with CCs isolated from the tracheal cartilage of the hyaline type is demonstrated to be a valid alternative to traditional healing for the treatment of articular cartilage lesions in horses, with possible extensions of similar treatments to other animals, most notably in humans. Further, clinical cases would strengthen this technique’s validity and help optimize and standardize the therapeutic protocol.

## Figures and Tables

**Figure 1 biomedicines-11-01602-f001:**
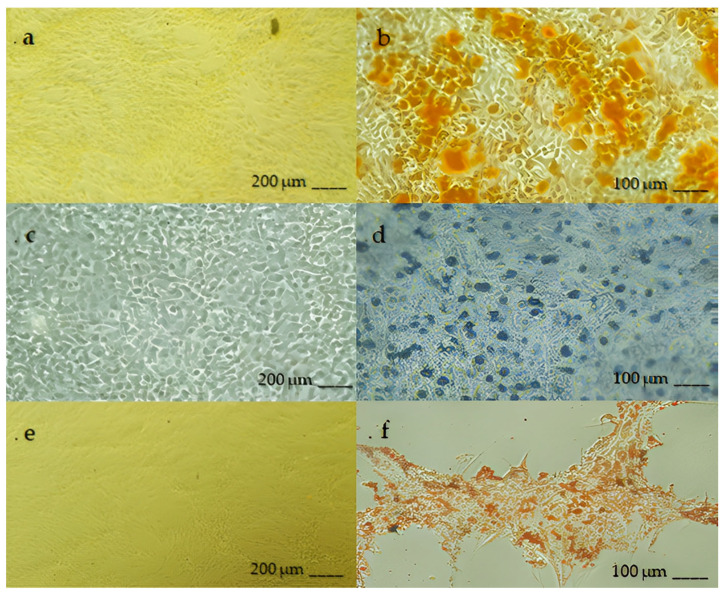
Optical microscopic images taken after in vitro differentiation of chondrocytes: (**a**) negative control of osteogenic differentiation (10×) staining with Alizarin Red S and (**b**) osteogenic differentiation (20×); (**c**) negative control of chondrogenic differentiation (10×) staining with Alcian Blue and (**d**) chondrogenic differentiation (20×); (**e**) negative control of adipogenic differentiation (10×) staining with Oil Red O and (**f**) adipogenic differentiation (20×).

**Figure 2 biomedicines-11-01602-f002:**
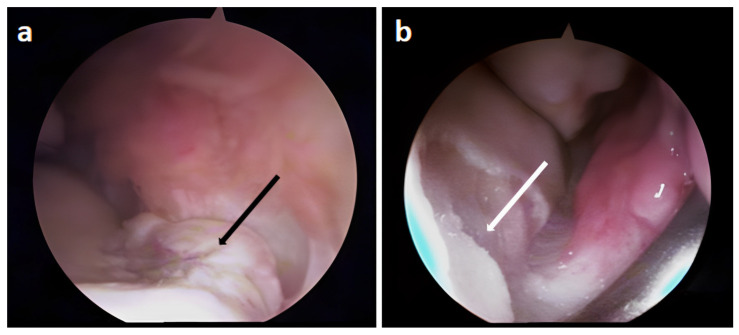
Intraoperative images of the first arthroscopy in which chondrocytes were inserted into the lesion site: (**a**) chondral lesion with evident fragment (arrow), (**b**) cartilage prepared for the inoculation of autologous chondrocytes after fragment removal and debridement of the surrounding area (arrow).

**Figure 3 biomedicines-11-01602-f003:**
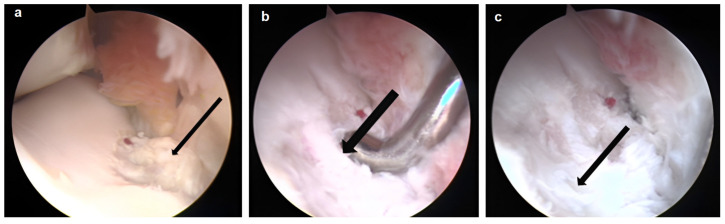
Intraoperative images of the second arthroscopy 8 months after the insertion of the chondrocytes at the lesion site: (**a**) evident portion of fibrocartilage at the site of the lesion adherent to the capsule (arrow), (**b**,**c**) more palmar with thinned cartilage and an island of tissue more similar to hyaline cartilage (indicated by the larger arrows).

**Figure 4 biomedicines-11-01602-f004:**
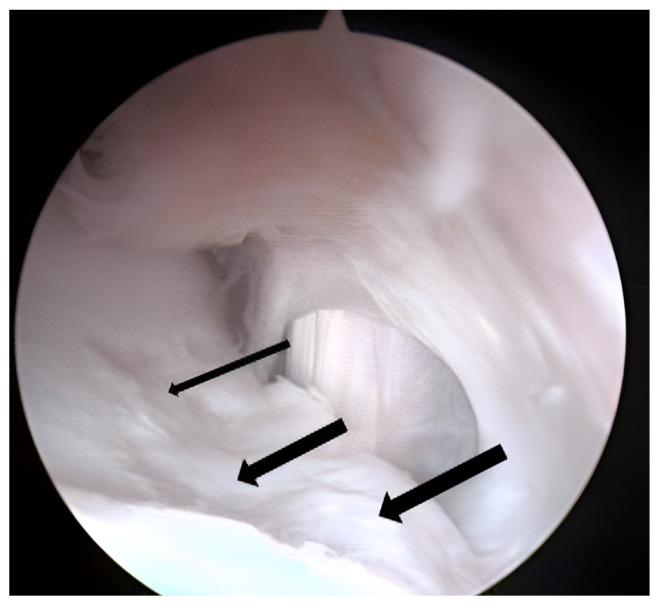
The arthroscopic image of the site treated 13 months after the inoculation of autologous chondrocytes inserted into the site of the lesion. Area of fibrocartilage was observed (thin arrow) and areas more comparable to hyaline cartilage were also present (thicker arrows).

**Figure 5 biomedicines-11-01602-f005:**
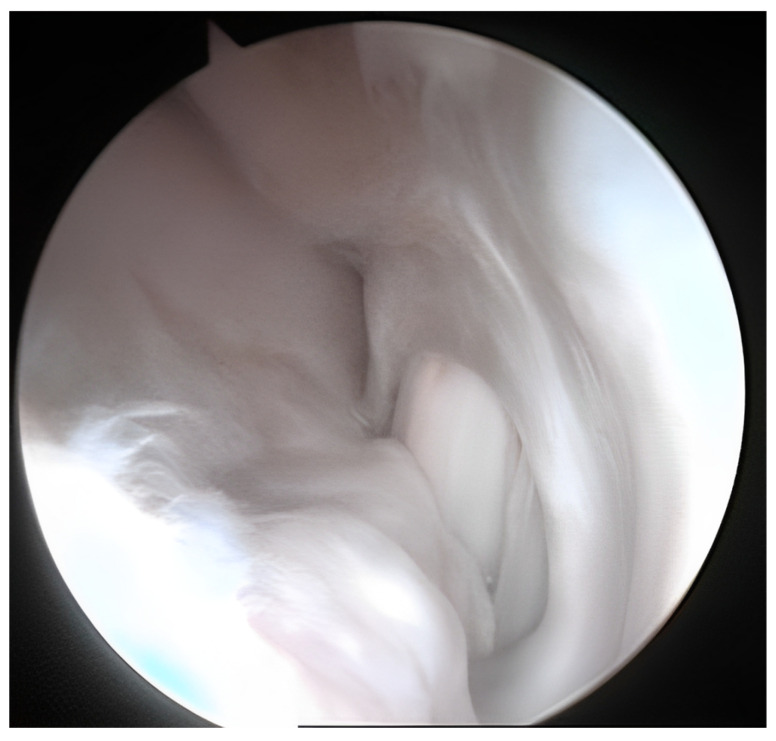
A final control was performed 24 months after surgery, and more extensive tissue comparable to hyaline cartilage was observed.

**Figure 6 biomedicines-11-01602-f006:**
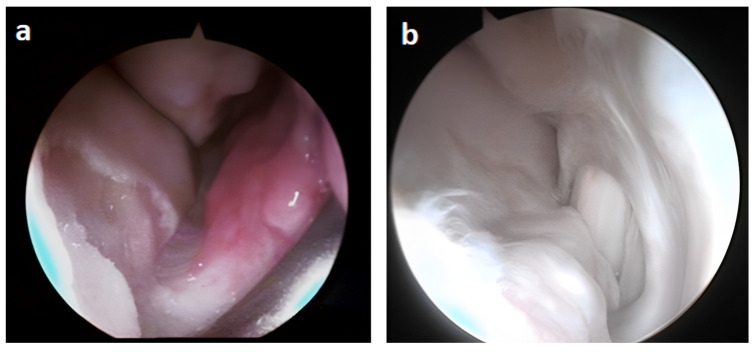
(**a**,**b**) The site of the lesion prepared for the implant of the chondrocytes at the first surgery (**a**) and the same site after 24 months (**b**).

**Figure 7 biomedicines-11-01602-f007:**
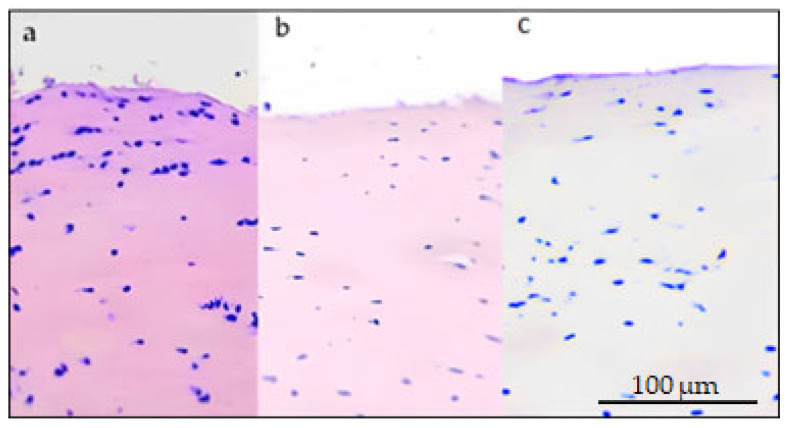
Articular bioptic sample, 10× magnification. (**a**) Treated lesion, 8 months post-surgery, hematoxylin eosin. Cluster of proliferating chondrocytes embedded in an eosinophilic, immature collagen matrix. (**b**) Treated lesion, 8 months post-surgery, IHC anti-COL2A1. There is no expression of collagen type II. (**c**) Untreated (control) lesion, 8 months post-surgery. IHC anti-COL2A1. There is no expression of collagen type II.

**Figure 8 biomedicines-11-01602-f008:**
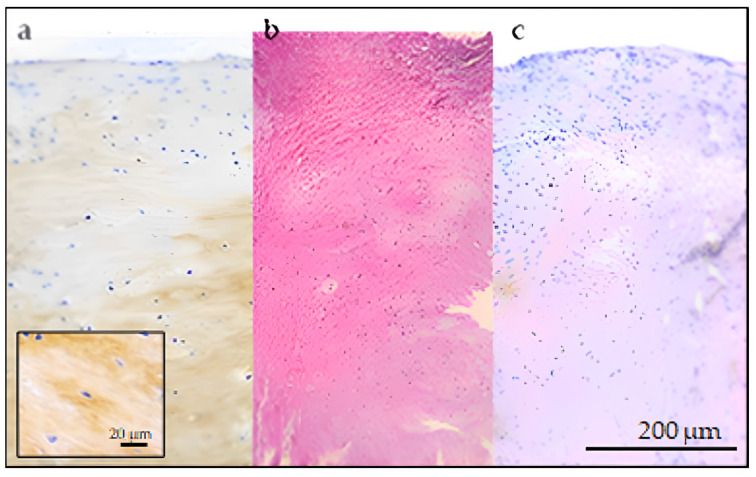
Articular bioptic sample, 5× magnification: (**a**) treated lesion, 13 months post-surgery IHC anti-COL2A1. There is patchy expression of collagen type II in the upper layer of proliferating chondrocytes, visible as brown discoloration in the site where antibodies reacted with Col2A1 antigen. (**b**) Untreated (control) lesion, 13 months post-surgery. Hematoxylin eosin. Fascicular pattern with intensely eosinophilic fibrous connective matrix. (**c**) Untreated (control) lesion, 13 months post-surgery. IHC anti-COL2A1. There is no expression of collagen type II.

**Figure 9 biomedicines-11-01602-f009:**
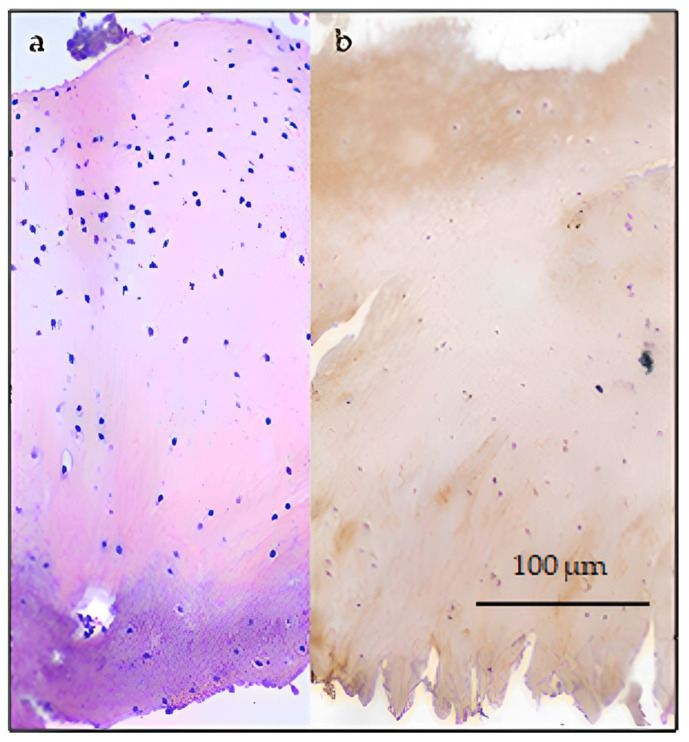
Articular bioptic sample, 10× magnification, treated lesion, 24 months post-surgery: (**a**) Hematoxylin eosin. Chondrocytes are morphologically arranged in a pattern mimicking the mature hyaline cartilage, (**b**) IHC anti-COL2A1. There is a diffuse expression of collagen type II up to the superficial aspects of the sample.

## Data Availability

Not applicable.

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
