# Peer review of "Articular Cartilage Regeneration by Hyaline Chondrocytes: A Case Study in Equine Model and Outcomes"

_biomedicines, 2023, doi:10.3390/biomedicines11061602_

Round 1

Reviewer 1 Report (Previous Reviewer 1)

Dear authors

Thanks for your answers and corrections. There are still a few details to correct.

Figure 3 : a, b, and c are missing.

Figure 7 hematoxylin instead of hematox-ylin, scale bars are missing. Please add high magnifications of the areas of interest to support the results.

Figure 8 scale bars are missing. Please add high magnifications of the areas of interest to support the results. Indicate precisely the staining in a.

Figure 9  scale bar is missing. Indicate precisely the staining in b.

Best regards

Author Response

Reviewer 1

Thanks for your answers and corrections. There are still a few details to correct.

Figure 3 : a, b, and c are missing. OK, we have included the symbols.

Figure 7 hematoxylin instead of hematox-ylin, OK for the error.

scale bars are missing. OK, we have included the scale bar.

Please add high magnifications of the areas of interest to support the results. The picture represents the morphology of the cartilage at 8 months post-surgery, we cannot identify neither visible differences between treated and untreated lesions, nor different expression of COL2A1; in this case we cannot highlight specific parts of the pictures and we believe that a higher magnification would show only few chondrocytes while we intended to show their pattern of proliferation.

Figure 8 scale bars are missing. Please add high magnifications of the areas of interest to support the results. Indicate precisely the staining in a.

We add an inset with magnification of the original picture to highlight the patchy expression of the anti-COL2A1 antibody, a scale bar of the inset is also provided.

The staining in Figure 8a is reported as IHC anti-COL2A1 that is Immunohistochemistry using an antibody that detect the antigen specific for the type 2 collagen molecule (as reported in the materials and methods section)

Figure 9 scale bar is missing. Indicate precisely the staining in b.

Scale bar is reported. The staining in Figure 9b is reported as IHC anti-COL2A1 that is Immunohistochemistry using an antibody that detect the antigen specific for the type 2 collagen molecule (as reported in the material and methods section)

Reviewer 2 Report (Previous Reviewer 2)

The authors have responded all the comments raised by the reviewer. The manuscript can be accepted in its present form.

Author Response

Thanks for your comments 

Best regards

Reviewer 3 Report (Previous Reviewer 3)

The authors have revised their manuscript in accordance with my comments that arose during the first round of revision. I have no other specific comments or questions and believe the manuscript can be accepted in its present form.

Author Response

Thanks for your comments 

Best regards

This manuscript is a resubmission of an earlier submission. The following is a list of the peer review reports and author responses from that submission.

Round 1

Reviewer 1 Report

Dear authors

I have a lot of comments and changes to ask you.

Please, specify the authorization number to experiment on horse

Detail if you use paraffin sections? Thickness of the sections…

You block peroxidase specify how.

Polyclonal instead of polyoclonal. Give the exact reference of the antibody used.

Why do you do the differentiation test? what is the point ? They aren not stem cells?

All figures to review:

- Figure 1 missing a, b, c, d. scale bar

- Figures 2 and 3 indicate the lesion with arrow, blurred images, increase quality, scale bar

- Figure 4 indicate lesion site and regeneration : fibrocartilage, hyaline cartilage. Show an image at 24 months and control for each age

- Figure 5 to be redone very ugly, we can't see anything.

- Figure 6 Redo this figure, please indicate where coll2A1 positive cells. Add scale bar. b) Crop to eliminate tears. 

Make magnifications to see the details: we do not see the structure of the cartilage. Stain with Alcian blue and/or safranin O. Show control cartilage to compare with that which is regenerated. Make other immuno with specific markers like aggrecan.

Results

Specify the size of the biopsy. The size sample does not affect regeneration?

3.1 it would be nice to show the cells in culture, their characterization by immuno, pcr, western blot.

Evaluation of differentiation to be put in the material and methods refer to the figures.

Problem of annotation of paragraphs 3 then 3.1 then 3.1.2

Reimage oil red. Normally the staining is red for the droplets, show an inset with droplets. Are all the photos at the same magnification?

Normally chondrogenic differentiation in 3D.

In 3.3 : Figure 5 and not 4 as written. Put more details of the cells, alcian blue and/or safranin O staining would be more specific, scale bar are missing.

Figure 6 make the same staining for all times, also put the controls for all times. Blurred photos.

Is there subchondral bone or only cartilage.

Best regards

Reviewer 2 Report

The Authors have performed a case study on chondrocyte cell transplantation to an equine cartilage defect model for analyzing its potential in cartilage defect regeneration. However, there have some concerns that I have with this manuscript at the present stage.

1.     The authors have evaluated tri-lineage differentiation potential of isolated chondrocytes by Alizarin Red, Oil Red O, and Alcian Blue staining. However, the authors need to add control with each differentiation study in Fig.1. Need more explanation regarding negative control and the staining.

2.     In Fig. 2, the chondrocyte insertion area and chondral lesion area should be marked.

3.     The authors need to include control in Fig. 5 for better understanding.

Reviewer 3 Report

The manuscript presented by Canonici et al. deals with a very relevant and practically important task: cartilage regeneration by autologous chondrocytes. The case study design is correct. The manuscript is well written and structured, and the results are supported by appropriate illustrative materials. The manuscript deserves to be accepted in Biomedicines, but I suggest a minor revision before publication.

1.     Introduction/Discussion

In my opinion, a discussion of similar studies or similar approaches used for cartilage reconstruction is desirable to emphasize the novelty of the described technique or to compare the results with similar studies.

2.     Section 2.2 and 3.2

Subsection 2.2.1/3.2.1 is highlighted in these sections. However, this separation does not seem to be necessary, as there are no other subsections in these sections. A single paragraph here would be more appropriate.